# Physicians' perspective on potentially non-beneficial treatment when assessing patients with advanced disease for ICU admission: a qualitative study

Monica Escher ®,[1,2] Mathieu R Nendaz,[2,3] Stéphane Cullati,[4,5] Patricia Hudelson[6]

► Prepublication history and additional onlne supplemental material for this paper are available online. To view these files, please visit the journal online (http://dx.doi.org/10.1136/bmjopen-2020-046268).

**Correspondence to**
Dr Monica Escher;
monica.escher@hcuge.ch

## ABSTRACT

**Objective** The use of intensive care at the end of life can be high, leading to inappropriate healthcare utilisation, and prolonged suffering for patients and families. The objective of the study was to determine which factors influence physicians' admission decisions in situations of potentially non-beneficial intensive care.

**Design** This is a secondary analysis of a qualitative study exploring the triage process. In-depth interviews were analysed using an inductive approach to thematic content analysis.

**Setting** Data were collected in a Swiss tertiary care centre between March and June 2013.

**Participants** 12 intensive care unit (ICU) physicians and 12 internists routinely involved in ICU admission decisions.

**Results** Physicians struggled to understand the request for intensive care for patients with advanced disease and full code status. Physicians considered patients' long-term vital and functional prognosis, but they also resorted to shortcuts, that is, a priori consensus about reasons for admitting a patient. Family pressure and unexpected critical events were determinants of admission to the ICU. Patient preferences, ICU physician's expertise and collaborative decision making facilitated refusal. Physicians were willing to admit a patient with advanced disease for a limited amount of time to fulfil a personal need.

**Conclusions** In situations of potentially non-beneficial intensive care, the influence of shortcuts or context-related factors suggests that practice variations and inappropriate admission decisions are likely to occur. Institutional guidelines and timely goals of care discussions with patients with advanced disease and their families could contribute to ensuring appropriate levels of care.

## Strengths and limitations of this study

► Participant sample was representative of physicians involved in intensive care unit (ICU) admission decisions in our institution.
► In-depth interviews were conducted by an experienced medical sociologist.
► Data analysis was done by a multidisciplinary research team including clinicians from the intensive care, internal medicine and palliative care fields, a medical sociologist and a medical anthropologist.
► The main limitation of this study is that it is a secondary analysis of interviews that did not specifically focus on the role of potentially non-beneficial treatment in ICU admission decisions.

## INTRODUCTION

The use of intensive care in the last month of life can be high, especially for non-cancer patients.[1] Providing non-beneficial treatments to patients with advanced disease only prolongs suffering at the end of life. It is associated with family distress[2] and healthcare staff burn-out.[3] Potentially non-beneficial interventions is a concern for patients cared for in intensive care units (ICU).[4 5] In 2015, several prominent professional societies, among which the American Thoracic Society and the European Society for Intensive Care Medicine, published a joint statement about how to respond to patients' or families' requests for potentially inappropriate treatments.[6] The term « potentially inappropriate » was recommended over « futile » since it was acknowledged that a patient's values and preferences can legitimately lead him or his family to request life-prolonging interventions when physicians consider those treatments to be inappropriate. The requested medical intervention must have some chance to achieve the patient's goal, and in this case the physicians' justifications for not providing it are ethically based. Of note, the statement does not give guidance about how to determine how much chance justifies to administer the requested treatment or to challenge its appropriateness. The definition of potentially inappropriate interventions was addressed in a subsequent statement of the Society of Critical Care Medicine.[5] The medical interventions should allow to achieve at least one of two goals: either the patient will be able to live outside the acute

care setting, or he will recover sufficient neurological function to perceive the treatment benefits. Physicians' clinical judgement, however, is central to the decision since they have to estimate survival and cognitive outcomes. Moreover, the guidance makes allowance for time-limited interventions that might promote a patient's goals of care.

The discussion about potentially non-beneficial interventions has mainly focused on the administration of treatments to patients staying in the ICU. However, it can be an issue during triage. For example, no consensus was found about limiting the admission to intensive care based on a patient's chances of survival, not even for a chance as low as 0.1% or less.[7] The lack of specific criteria for ICU admission decisions has been recently pointed out.[8] Whereas a decision supporting framework was developed to address the issue of limiting or not life-sustaining treatments in the ICU,[9] no such framework exists for triage. Yet, deciding whether to admit a patient to the ICU is often complex, and physicians mostly rely on their clinical judgement.[10]

As significant knowledge gaps remain about the provision of potentially non-beneficial care, more studies on the topic have been called for.[11] Triage is an important area in this respect. Numerous patient-related and context-related factors were shown to influence the decision to admit or not a patient to the ICU, but data are lacking about how these various factors come into play within the decision making process.[12 13] When physicians assess a critically ill patient for admission to the ICU, they evaluate the medical indication—that is, added benefit of intensive care in terms of short-term prognosis—on the one hand, and long-term survival, potential for functional and cognitive recovery, and patient preferences on the other hand. The latter factors are framed in terms of goals of care. Based on their assessment, physicians determine what they think is appropriate treatment intensity for the patient. A particularly difficult situation involves critically ill patients with advanced disease for whom physicians consider limiting treatment intensity, and who have a full code status.[14] Although physicians have no obligation to follow a patient's code status, they cannot disregard it lightly. It is a strong indicator of treatment intensity, intended to guide decisions in case of an unexpected critical event. Therefore, to go against code status, that is, not to admit a patient to the ICU, is a difficult decision to make. We aimed to determine which factors physicians consider when they are faced with the ethical issue of providing potentially non-beneficial intensive care to a critically ill patient, and how these factors influence ICU admission decisions.

## METHODS

This is a secondary analysis of a qualitative study exploring the triage process.[14] The study was conducted at a tertiary care hospital.

### Participants and data collection

Physicians working in the Divisions of General Internal Medicine and of Intensive Care, and routinely assessing patients for intensive care were eligible. We included physicians from the two specialties because triage is a collaborative process in our institution. The internal medicine physician gives the ICU physician the relevant clinical information and goals of care of the critically ill patient. The ICU physician personally evaluates the patient and gives expert advice. The two physicians discuss whether or not to admit the patient to intensive care, but the ICU physician usually has the final say.

We used a combination of convenience and snowball sampling, and included equal numbers of internists (n=12) and ICU physicians (n=12). Study participants were representative of the physicians who make ICU admission decisions in our institution. Internists included both certified chief residents (n=8) and residents (n=4), since the latter are involved in admission decisions during night calls. ICU physicians were chief residents (n=7) and attendings (n=5). Participants were recruited between March and June 2013 after we presented the study at staff meetings and through email invitations. Interested physicians contacted one of the researchers (SC). At the end of the interview they were asked whether they knew of a colleague who might participate. All the identified physicians accepted to be included in the study. The participants gave written consent to participate in the study.

The interview guide was pretested with two internists and two ICU physicians (online supplemental file). A male PhD medical sociologist (SC) conducted face-to-face in-depth interviews. He was a member of the research team, had extensive experience in qualitative research and an interest in interprofessional collaboration and sociology of healthcare professions. He had neither previous nor hierarchical relationships to the interviewees. All interviews took place at the hospital, in a dedicated room outside the Division of General Internal Medicine and the Division of Intensive Care. SC introduced himself to participants as a sociologist collaborating on the research project.

Participants were invited to reflect on their experience of two ICU admission decisions involving a medical inpatient. They were asked to choose significant cases with regard to the way the decision was made. We indicated that the decision itself—admission or no admission—was not important, that the decision making process could have gone either smoothly or not, and that the clinical situations could be simple or complex. During the interviews the participants sometimes freely referred to other clinical situations, either to make their point or to expand on the idea they were developing. The main objective of the study was to identify the factors that facilitated or hindered admission decisions. Interviews lasted 57 min on average (min 26, max 94). They were recorded, transcribed verbatim and anonymised. No field notes were taken and each physician was interviewed only once. Participants were not asked to read and react to

the transcripts of their interviews, but three ICU physicians and two internists were presented with the main results of the study and asked whether they reflected their experiences.[14]

## Analysis

Interview transcripts were analysed using an inductive approach to thematic content analysis.[15] This approach enables to identify meaningful information regarding the research question from the textual data, and to relate it to overarching themes. Themes are analysed and interpreted into a coherent descriptive model. Analysis aimed to identify factors that influenced participants' decision-making around ICU admission. In particular, we were interested in understanding participants' views regarding potentially non-beneficial intensive care ('medical futility').

Four interviews (two with internists, two with ICU physicians) were first independently read and then discussed by members of the research team (ME, SC, MRN and PH). Based on this first reading, a preliminary list of codes was developed, and independently applied by SC and ME to the same four interviews. Any coding discrepancies were resolved by consensus, and a third researcher (PH or MRN) cross-checked the coded interviews. A finalised codelist was then applied by ME or SC to the remaining interviews, and then codings were cross-checked by two researchers (ME or SC, and PH or MRN). Whenever new ideas appeared in the interviews, new codes were created and then applied to all interviews. Codes were clustered according to their content relatedness (eg, intensive care

as default option'). Coding and analysis were conducted using Atlas.ti Scientific Software Development (V.7.0.71).

## Patient and public involvement

No patients were involved.

## RESULTS

### Participant characteristics

Among the 24 physicians, 17 were male. Mean age was 38 years (range 27–51) and mean number of years since graduation was 11.8 (SD 6.8). On average ICU physicians were older and more experienced than internists. Three internal medicine residents had never worked in an ICU. Participants' characteristics reflected medical staff's training background and working organisation in our institution. Most ICU physicians train in a primary specialty before training in critical care medicine. Since only senior ICU physicians evaluate critically ill patients on the wards, the differences in age and experience between intensivist and internist participants were expected.

### Clinical situations during triage

Physicians described two scenarios, when the decision to admit or refuse a patient to intensive care was straightforward (figure 1). Either there was a medical indication, that is, short-term benefit, and high intensity care was considered appropriate and was congruent with code status, then the patient was admitted; or there was no medical indication, and then the patient was refused.

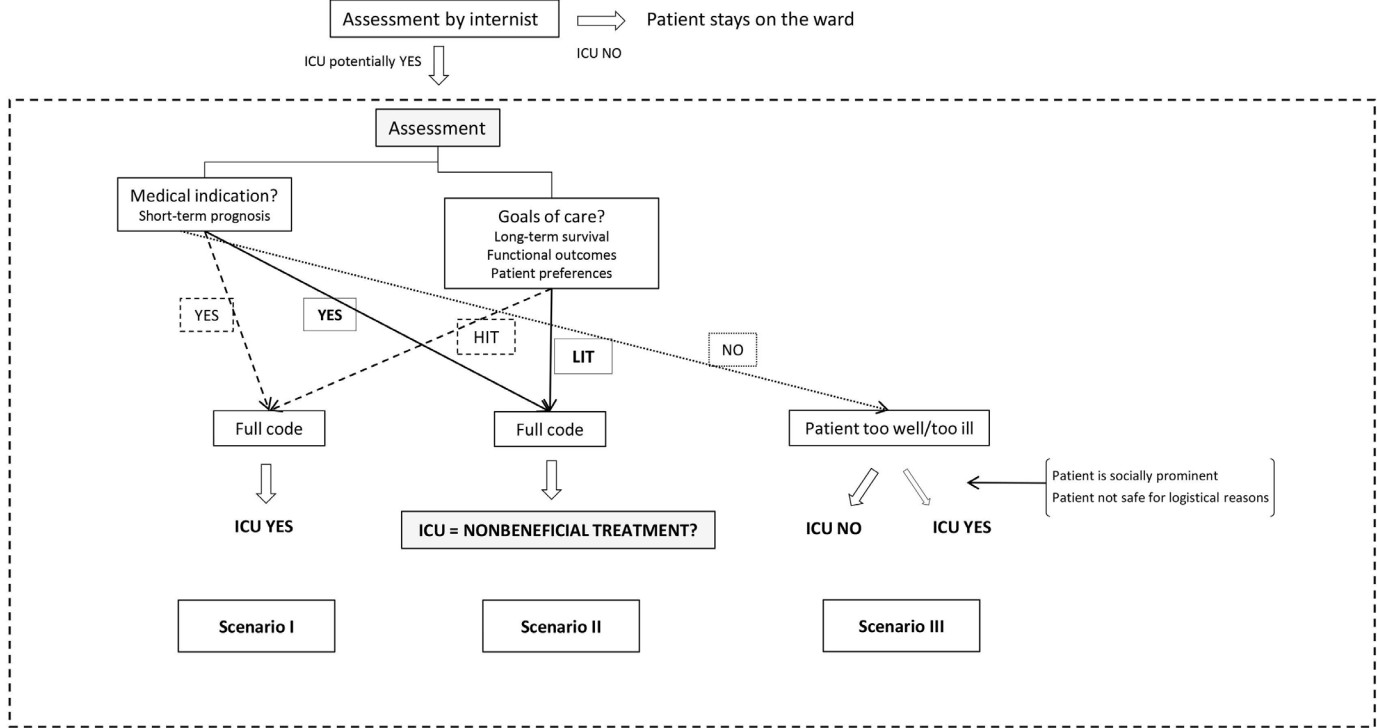

**Figure 1** Triage to intensive care: decision-making scenarios. [•••] → determinant; ▭⇨ decision; – → composed of. ICU, intensive care unit; HIT, high intensity treatment; LIT, low intensity treatment.

In situations where there was no medical indication, physicians explained that sometimes context-related factors, that is, social pressure due to a patient's prominence, and concerns about patient's safety on the ward, could lead to the patient being admitted.

> Probably the patient would not have had any benefit from intensive care, … but sometimes we must admit [a patient], precisely when there is some doubt, because we choose the safe side. (ICU12)

Participants explicitly raised the issue of potentially non-beneficial intensive care when they reported being faced with a dilemma. The dilemma arose from the discrepancy between their assessment—low intensity care more appropriate—and the high intensity care required by a full code status. It usually concerned patients with advanced disease as these patients could benefit from life-sustaining interventions, but their long-term survival prognosis and their capacities for cognitive and functional recovery were limited. (scenario II). In these situations, physicians struggled to make sense of the request for treatment.

> When the patient has a cancer at a very advanced stage and still, it is decided to intubate him because he has a pulmonary infection, is an admission to intensive care really meaningful? (ICU01)

### Factors influencing ICU admission decision in the case of potentially non-beneficial treatments

Participants described factors that oriented a decision towards admission, towards refusal or that were used for either decision (figure 2). There was consensus among respondents that intensive care should be provided as a default option in cases of great uncertainty, for patients needing intensive care as a consequence of an iatrogenic event, or for patients with onco-haematological diseases.

> When in doubt, we admit and we treat. (ICU10)

> When there are so-called iatrogenic complications, I feel I have a responsibility to treat the complication, … to make abstraction of the patient's general context and to use all available means to take care of it. (MED11)

In addition, respondents reported that they could be pressured into admitting a patient with advanced disease by the family or the referring physician. Factors related to the acute event could also prompt physicians to admit a patient for whom limited treatment intensity had previously been decided.

> Some families demand everything, even though it is futile,… and they put an enormous pressure on the system. (ICU04)

Physicians were also willing to provide life-sustaining treatments to a terminally ill patient for a limited amount of time in order to fulfil a personal need of the patient or family.

> Even in a desperate situation, we can admit a patient to intensive care if we know there is something coming up; we wait for a relative who is on his way (ICU11)

Determinants of ICU refusal in the case of potentially non-beneficial treatment involved not only consideration of patient preferences but were also influenced by professional interactions. The ICU physician's expertise carried

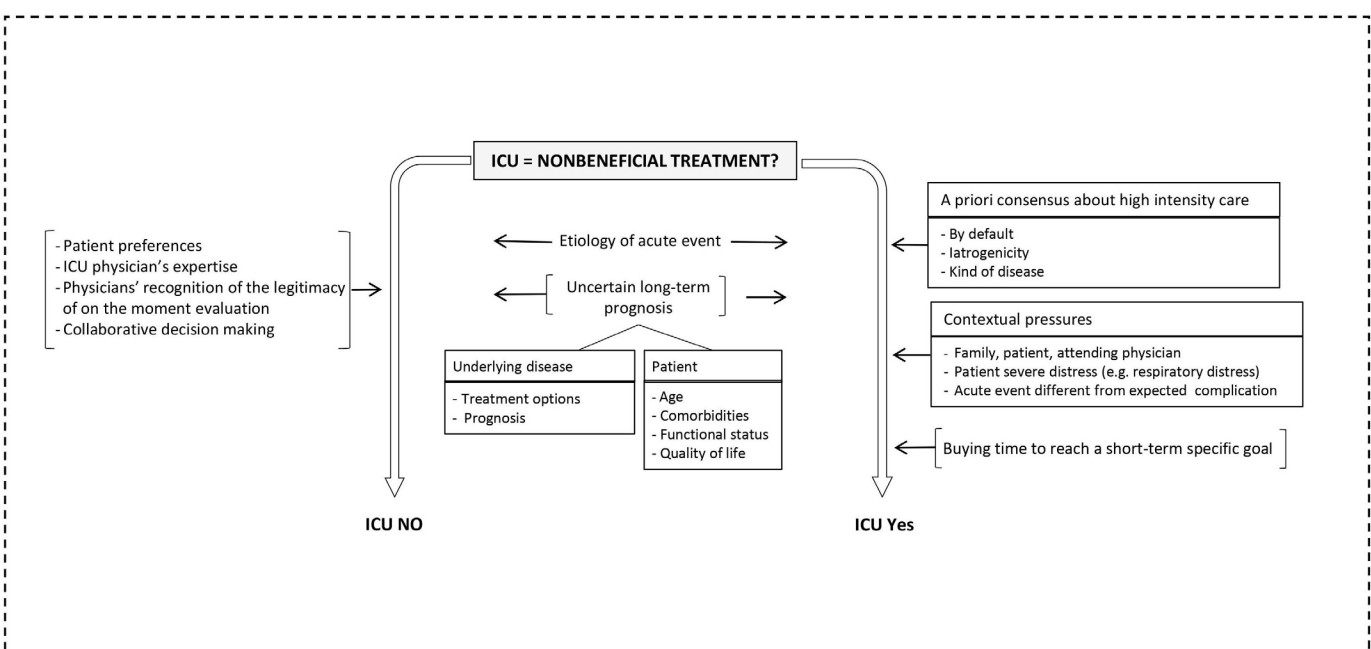

**Figure 2** Scenario II: factors influencing the decision towards admission to or refusal of intensive care. [●●●] → determinant; ⇨ decision; − → composed of. ICU, intensive care unit.

weight; collaborative decision making between internists and ICU physicians facilitated refusal as did physicians' recognition that ad hoc evaluation was at times as valuable as code status.

> We decided to go against the code status. But we did it together, we evaluated the patient, we discussed (MED01)

In such cases, and depending on the type of the acute event, the patient could be admitted or not. Physicians also took into account long-term prognosis. They considered patient-related factors, that is, age, comorbidities, functional status and quality of life and disease-related factors, that is, prognosis and availability of disease-directed treatments

## DISCUSSION

Physicians in our study explicitly integrated the provision of potentially non-beneficial treatment into the decision-making process of ICU admission when they were faced with a dilemma. The dilemma concerned patients with advanced disease who were full code, but for whom physicians considered low intensity care to be more appropriate. In these situations physicians took many factors into account, which reflects how complex the decision may be. They reasoned about patient's long-term prognosis, but they also resorted to shortcuts, that is, a priori consensus about reasons for admitting a patient. Human factors influenced the decision towards admission: physicians felt pressure on the part of the family and as a consequence of unexpected critical events. More positively, physicians were willing to admit a patient if it could enable him to reach a meaningful short-term goal. Professional factors facilitated the decision towards refusal of intensive care: medical expertise, in particular the ICU physician's and collaborative decision making.

Our findings show that the provision of potentially non-beneficial treatments can be an issue for physicians during triage as it is in the ICU.[4] The determinants of ICU admission or refusal in these situations are on the whole similar to the ones reported in the current literature about the general process of decision making for ICU admissions.[12 13] Physicians consider longer-term survival and functional outcomes, and are influenced by patient preferences and context-related factors.

The use of short-cuts in admission decisions contrasts with the process advocated to decide about limiting or not life-sustaining treatments in the ICU.[9] It reflects the time-pressured context of triage when repeated meetings with the family and among the healthcare professionals are hardly feasible and when prognostic uncertainty is high. To admit a patient in case of great uncertainty is consistent with professional guidelines that deem overtriage to be more acceptable than undertriage.[11] Admission when in doubt is a behaviour physicians reported previously.[16] Physicians' response to unexpected events could be ethically problematic and lead to potentially inappropriate admissions to intensive care. It is likely that patients' perspectives differ from physicians' in this respect as a recent study has shown

that patients are willing to trade survival time to avoid end of life in an ICU.[17] Family opinion has been shown to significantly influence ICU admission decisions.[14 18 19] Family can either act as useful healthcare surrogates or make requests in response to their own needs.[20 21] Interestingly physicians referred to family only as putting pressure towards admission in situations of potentially non-beneficial treatments. It epitomises the difficulty of responding to such requests, which has prompted the issuance of guidance by professional societies.[6] Disagreement between medical team and family has been associated with perceived inappropriate care in the ICU,[22] and also with potentially inappropriate admissions to the ICU from hospital wards.[23] Similarly to our study, pressure from the referring physician to provide potentially non-beneficial treatments has been reported in the ICU setting.[22]

ICU physicians' expertise and collaborative decision making are factors that can facilitate a decision not to admit a patient. Clinician experience was also found to have a significant influence on challenging ICU admission decisions in a qualitative study about triage in the emergency department.[24] Such decisions are difficult to make and physicians' willingness to admit patients to the ICU so that they or their family could fulfil a personal need is in keeping with current attitudes. Time-limited trial is an accepted strategy for patients with a poor prognosis when survival benefit with intensive care or patient preferences are unclear, or when patient and/or family need time to adapt.[25] Such an approach is concordant with the intention to provide patient-centred and family-centred sensitive care.

Our study has limitations. It is a secondary analysis of interviews that did not specifically focus on the role of potentially non-beneficial treatment in ICU admission decision making. Other issues might arise in a more in-depth study on this topic. In addition, the study was conducted in a context where internists and ICU physicians collaborate when deciding on ICU admission. Where this is not the case, physicians may be influenced by different factors. Nonetheless, data about triage and the provision of potentially non-beneficial treatments are scarce and our study brings novel insights into physicians' decision making under these time-pressured circumstances. We were able to identify several patient-related, physician-related and context-related factors, and we could determine in which direction these various factors influenced the ICU admission decision.

## CONCLUSION

Physicians are concerned about providing potentially non-beneficial intensive care treatment for critically ill patients with advanced disease in situations of uncertainty. The ICU admission decision is then complex and influenced by a variety of medical and contextual factors. The role that shortcuts or context-related factors may play raises concerns about potentially inappropriate admission to intensive care. Our results highlight the risk of practice variation in ICU admission decisions. Additional research should focus on

how physicians weigh multiple contextual factors, and on how institutional guidelines and advance care planning with patients and families can help admission decisions and contribute to ensuring appropriate levels of care.

**Author affiliations**
[1]Division of Palliative Medicine, University Hospitals of Geneva, Geneva, Switzerland
[2]Unit for Development and Research in Medical Education (UDREM), Faculty of Medicine, University of Geneva, Geneva, Switzerland
[3]Division of General Internal Medicine, University Hospitals of Geneva, Geneva, Switzerland
[4]Quality of Care Service, University Hospitals of Geneva, Geneva, Switzerland
[5]Population Health Laboratory, Faculty of Science and Medicine, University of Fribourg, Fribourg, Switzerland
[6]Department of Primary Care, University Hospitals of Geneva, Geneva, Switzerland

**Acknowledgements** The authors thank the physicians who participated in the study. We also thank Prof Thomas Perneger for his invaluable contribution to study concepts and design, and Prof Bara Ricou for her contribution to the whole project about the determination of factors influencing ICU admission decisions.

**Contributors** ME contributed to study concepts. ME, MRN, SC and PH contributed to study design. SC and ME collected the data. All the authors contributed to quality control of the data. All authors contributed to data analysis and interpretation. ME drafted the manuscript. All authors contributed to manuscript editing and review.

**Funding** This work was supported by the Swiss National Science Foundation, National Research Program 'End of Life' (NRP 67) grant number 139304.

**Competing interests** None declared.

**Patient consent for publication** Not required.

**Ethics approval** This study was approved by the Geneva Research Ethics Committee (12-042).

**Provenance and peer review** Not commissioned; externally peer reviewed.

**Data availability statement** All data relevant to the study are included in the article or uploaded as online supplemental information. No additional data are available.

**ORCID iD**
Monica Escher http://orcid.org/0000-0002-7167-4550

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
