## [Reviewer comments · BMJ Open]

ARTICLE DETAILS

TITLE (PROVISIONAL)	Physicians' perspective on potentially nonbeneficial treatment when assessing patients with advanced disease for ICU admission: a qualitative study.
AUTHORS	Escher, Monica; Nendaz, Mathieu; Cullati, Stéphane; Hudelson, Patricia

VERSION 1 – REVIEW

REVIEWER	Simon Oczkowski McMaster University, Canada
REVIEW RETURNED	26-Nov-2020

GENERAL COMMENTS	Thank you for the opportunity to review this interesting paper. Despite being a secondary analysis it provides valuable insights into the decision-making process for ICU admission. There are a few areas which I think would benefit from clarification: 1) What is the medicolegal context of these clinicians? Are they obligated to provide ICU admission, or can they over-ride a patient's code status decisions? This is important to understand as in some cases the physicians may not actually have a choice of going against patient and family wishes. The quote by MED01 on page 11 implies the physicians can actually go against code status. 2) It would be helpful to provide some potential definitions for non-beneficial treatment. It is admittedly a challenge, but a major aspect of the work is to identify how clinicians make decisions about admission to ICU nonbeneficial treatment. Is it treatment unlikely to prolong survival to hospital discharge? Treatment for patients who are unlikely to be conscious enough to experience benefit etc? 3) There is a bit of an inconsistency in the model developed, which may be due to an inconsistency in the way clinicians describe their decision-making process, or it may be an inconsistency that the investigators may choose to address and reconsider the scheme. The box labeled "TI = code status" indicates that clinicians consider that treatment may be non-beneficial when treatment intensity is not concordant with code status. But, how is this decision made? In the framework, decisions about whether a treatment is non-beneficial come later. What are the flags that suggest to clinicians that treatment is non-beneficial? I would imagine that these indicators come quite early in the process. As written the schema seems to indicate "when goals of care are not concordant with treatment intensity, clinicians consider that the treatment may be non-beneficial" which is begging the question (it
---

	is the fact that treatment is non-beneficial is why treatment intensity is not concordant with code status). 4) How were clinicians prompted to choose the two significant cases? Were they asked to choose typical cases, challenging cases, etc? 5) It is not clear to me how the schema was developed-- was this a grounded theory approach?
--	--

REVIEWER	Yong Liu ICU ,Shenzhen hospital, Southern medical university
REVIEW RETURNED	27-Dec-2020

GENERAL COMMENTS	This article addresses an important issue, both to readers in developed and developing countries. The study's aim was worthwhile, and the chosen methodology was appropriate. Manuscript is well written. A few recommendations as bellow:  - Please summarize and evaluate the findings from previous, relevant studies in the introduction. In addition, please describe the knowledge gap that you are addressing and provide the value/significance of your research despite of the existing studies. - Several keys references should be included but not limited to:  □ Engdahl Mtango S, Lugazia E, Baker U, Johansson Y, Baker (2019) Referral and admission to intensive care: A qualitative study of doctors' practices in a Tanzanian university hospital. PLoS ONE 14(10): e0224355. https://doi.org/10.1371/journal.pone.0224355. □ Anke undertook an explorative, descriptive study using qualitative methods (individual and focus group interviews) in the ICU or the general ward of 10 Dutch hospitals and provide insight into ethical problems that influence the ICU admission and discharge process - The result part is not informative enough for an in-depth interview, even for a qualitative study. More information should provide, such as: Is there any reason why the physician role of internists and ICU physicians were different? Chief residents and attendings in ICU group have longer training duration than certified chief residents and residents in internists participants. - Pls compare your findings to the mainstream and latest studies relevant to your research. I have suggested some ref. and pls make further and focused search.
--

REVIEWER	Alexandra Papaioannou University Hospital of Heraklion Greece
REVIEW RETURNED	30-Dec-2020

GENERAL COMMENTS	It is an interesting study, addressing issues all ICU doctors face. However, in methods, (and since it is a secondary analysis) it is not clear whether the results presented here reflect only the cases where the participants considered that ICU admission was futile. Participants were asked to reflect on 2 patients but it is not clear how many those wouldn't benefit from ICU treatment
---

VERSION 1 – AUTHOR RESPONSE

Reviewer: 1
Dr. Simon Oczkowski, McMaster University

Thank you for the opportunity to review this interesting paper. Despite being a secondary analysis it provides valuable insights into the decision-making process for ICU admission. There are a few areas which I think would benefit from clarification:

1) What is the medicolegal context of these clinicians? Are they obligated to provide ICU admission, or can they over-ride a patient's code status decisions? This is important to understand as in some cases the physicians may not actually have a choice of going against patient and family wishes. The quote by MED01 on page 11 implies the physicians can actually go against code status.

Thank you for the question. Clinicians have no obligation to follow code status. It is a strong indication of treatment intensity, but code status can be overridden if the clinicians think intensive care is inappropriate after they have assessed a critically ill patient. However, to go against code status, i.e. not to admit the patient to the ICU, is a difficult decision to make.

We clarified the context in the Introduction section.

pp.7-8, lines 45-48 : Although physicians have no obligation to follow a patient's code status, they cannot disregard it lightly. It is a strong indicator of treatment intensity, intended to guide decisions in case of an unexpected critical event. Therefore, to go against code status, i.e. not to admit a patient to the ICU, is a difficult decision to make.

2) It would be helpful to provide some potential definitions for non beneficial treatment. It is admittedly a challenge, but a major aspect of the work is to identify how clinicians make decisions about admission to ICU nonbeneficial treatment. Is it treatment unlikely to prolong survival to hospital discharge? Treatment for patients who are unlikely to be conscious enough to experience benefit etc? The reviewer is quite right. Defining nonbeneficial treatment is no simple matter. It is one reason why the term "potentially" nonbeneficial interventions is more and more used. There have been attempts to find a common understanding and a definition of potentially nonbeneficial treatment (Kon AA, et al. Crit Care Med 2016; DOI: 10.1097/CCM.0000000000001965. Reference # 5).

We gave information about the definition of nonbeneficial intensive care treatment in the Introduction section. We also added the word "potentially" when appropriate, wherever we hadn't used it in the manuscript, including in the title.

p. 6, lines 6-23: In 2015 several prominent professional societies, among which the American Thoracic Society and the European Society for Intensive Care Medicine, published a joint statement about how to respond to patients' or families' requests for potentially inappropriate treatments in the intensive care unit. (Bosslet GT, et al. Am J Respir Crit Care Med, 2015) The term « potentially inappropriate » was recommended over « futile » since it was acknowledged that a patient's values and preferences can legitimately lead him or his family to request life-prolonging interventions when physicians consider those treatments to be inappropriate. The requested medical intervention must have some chance to achieve the patient's goal, and in this case the physicians' justifications for not providing it are ethically based. Of note, the statement does not give guidance about how to determine how much chance justifies to administer the requested treatment, or to challenge its appropriateness. The definition of potentially inappropriate interventions was addressed in a subsequent statement of the Society of Critical Care Medicine. (Kon AA, et al. 2016) The medical interventions should allow to achieve at least one of two goals: either the patient will be able to live outside the acute care setting, or he will recover sufficient neurologic function to perceive the treatment benefits. Physicians' clinical judgment however is central to the decision since they have to estimate survival and cognitive outcomes. Moreover, the guidance makes allowance for time-limited interventions that might promote a patient's goals of care.

3) There is a bit of an inconsistency in the model developed, which may be due an inconsistency in the way clinicians describe their decision-making process, or it may be an inconsistency the investigators may choose to address and reconsider the scheme. The box labeled "TI = code status" indicates that clinicians consider that treatment may be non beneficial when treatment intensity is not concordant with code status. But, how is this decision made? In the framework, decisions about whether a treatment is non beneficial come later. What are the flags that suggest to clinicians that treatment is non beneficial? I would imagine that these indicators come quite early in the process. As

written the schema seems to indicate "when goals of care are not concordant with treatment intensity, clinicians consider that the treatment may be non-beneficial" which is begging the question (it is the fact that treatment is non-beneficial is why treatment intensity is not concordant with code status). We thank the reviewer for pointing out this apparent inconsistency. We agree that the figure synthesizing the model can be misleading.

To determine whether patients can benefit from intensive care is a routine task during triage. When physicians assess a critically ill patient for admission to the ICU, they evaluate the medical indication – i.e. added benefit of intensive care in terms of short-term prognosis - on the one hand, and long-term survival, potential for functional and cognitive recovery, and patient preferences on the other hand. The latter factors are framed in terms of goals of care. Based on their assessment, physicians determine what they think is appropriate treatment intensity for the patient. It is a stepwise decision in the sense that the absence of medical indication will most of the time overrides any other consideration. The patient is considered either too well or too ill to benefit from intensive care (scenario III in figure 1, revised version of the manuscript), and will usually not be admitted. In the interviews, participants explicitly raised the issue of potentially nonbeneficial intensive care when they reported being faced with a dilemma. The dilemma arose from the discrepancy between their assessment – low intensity care more appropriate – and the high intensity care required by a full code status. It usually concerned patients with advanced disease as these patients could benefit from life-sustaining interventions, but their long-term survival prognosis and capacities for functional recovery were limited. This is the focus of our study. It explores the factors physicians consider to solve the dilemma and how these factors influence the decision, either towards ICU admission or towards refusal.

We completed the Introduction and the Results sections, and thoroughly revised the figure illustrating the model in order to clarify the data and their meaning. We split the model into 2 figures: one illustrating the decision making process according to the 3 scenarios, and the other illustrating how the various factors influence the admission decisions when physicians explicitly identify the issue of providing potentially inappropriate intensive care.

p.7, lines 37-43: When physicians assess a critically ill patient for admission to the ICU, they evaluate the medical indication – i.e. added benefit of intensive care in terms of short-term prognosis - on the one hand, and long-term survival, potential for functional and cognitive recovery, and patient preferences on the other hand. The latter factors are framed in terms of goals of care. Based on their assessment, physicians determine what they think is appropriate treatment intensity for the patient.

pp.13-14, lines 147-152: Participants explicitly raised the issue of potentially nonbeneficial intensive care when they reported being faced with a dilemma. The dilemma arose from the discrepancy between their assessment – low intensity care more appropriate – and the high intensity care required by a full code status. It usually concerned patients with advanced disease as these patients could benefit from life-sustaining interventions, but their long-term survival prognosis and their capacities for cognitive and functional recovery were limited.

p.14, line 155: In these situations, physicians struggled to make sense of the request for treatment
4) How were clinicians prompted to choose the two significant cases? Were they asked to choose typical cases, challenging cases, etc?

The primary aim of the study was to explore the decision making process of ICU admissions and to determine the factors influencing the decisions. We asked clinicians to choose 2 cases in which they were personally involved and which they found significant regarding the way the decision was made. We clarified that the decision itself – admission or no admission – was not important. We also indicated that they could describe situations when the decision making process either went well or not, and that the clinical situations could be simple or complex.

We gave more details about the way physicians were prompted to choose two cases in the Methods section.

p.10, lines 87-90 : They were asked to choose significant cases with regard to the way the decision was made. We indicated that the decision itself – admission or no admission – was not important, that

the decision making process could have gone either smoothly or not, and that the clinical situations could be simple or complex.

5) It is not clear to me how the schema was developed-- was this a grounded theory approach?

We used thematic analysis (see also ref. 8: Hsieh Hf, Shannon SE, 2005), and not a grounded theory approach. Thematic analysis is an inductive approach used to identify themes from the textual data. A theme consists in an overarching descriptive analysis of meaningful information emerging from the data. The methodology is based on several major steps: independent reading of the interviews by several researchers and identification of meaningful information with regards to the research question (e.g. iatrogenicity, ICU admission as a default option); development of a list of codes that is used to code further interviews and that is continuously refined until no new code (i.e idea) emerges; clustering of codes into a theme according to content relatedness (e.g. a priori consensus about high intensity care); and analysis and interpretation of the themes into a coherent descriptive model. This goal differs from grounded theory since it is not an attempt at developing and testing a theory by means of iterative collection of data.

We specified the particulars of a thematic analysis approach in the Methods section.

p.11, lines 102-104: This approach enables to identify meaningful information regarding the research question from the textual data, and to relate it to overarching themes. Themes are analysed and interpreted into a coherent descriptive model.

Reviewer: 2

Dr. Yong Liu, Shenzhen Hospital, Southern Medical University

This article addresses an important issue, both to readers in developed and developing countries. The study's aim was worthwhile, and the chosen methodology was appropriate. Manuscript is well written.

A few recommendations as below:

1) Please summarize and evaluate the findings from previous, relevant studies in the introduction. In addition, please describe the knowledge gap that you are addressing and provide the value/significance of your research despite of the existing studies.

Several keys references should be included but not limited to:

Engdahl Mtango S, Lugazia E, Baker U, Johansson Y, Baker (2019) Referral and admission to intensive care: A qualitative study of doctors' practices in a Tanzanian university hospital. PLoS ONE 14(10): e0224355. <https://doi.org/10.1371/journal.pone.0224355>.

Anke undertook an explorative, descriptive study using qualitative methods (individual and focus group interviews) in the ICU or the general ward of 10 Dutch hospitals and provide insight into ethical problems that influence the ICU admission and discharge process

We thank the reviewer for his comment and the references. The articles mentioned deal with the factors influencing the admission to the ICU in general. This important topic was a primary objective of our qualitative study, with the exploration of the ICU admission decision making process. The results have been published and are referenced in this manuscript (reference 7: Escher M, et al. Health Services Research, 2019; doi: 10.1111/1475-6773.13076). In our previous publication we refer to the work of A. Oerlemans et al, as well as to other relevant studies.

In this secondary analysis, we focused on situations when participants explicitly express being faced with a dilemma, i.e. admitting or not a patient whose critical illness can justify life-sustaining interventions and who is full code, but for whom physicians consider intensive care to be inappropriate because of unfavorable longer term survival and functional outcomes. We aimed to determine how physicians reach a decision under these specific circumstances: which factors they consider and how these factors influence the decision either towards admission to the ICU or towards refusal. Research has been conducted about the provision of potentially nonbeneficial medical interventions for patients cared for in ICUs. However, there remain important knowledge gaps and more studies on the topic have been called for (Nates JL, et al. Crit Care Med 2016). Triage is still an underresearched area in this respect.

We described the focus of our study and the knowledge gap it addressed more precisely in the Introduction section, and added the relevant references, including the study by A. Oerlemans et al. p.7, line 24-30: The discussion about potentially nonbeneficial interventions has mainly focused on the administration of treatments to patients staying in the ICU. However it can be an issue during triage. For example, no consensus was found about limiting the admission to intensive care based on a patient's chances of survival, not even for a chance as low as 0.1% or less.(Sprung CL, Danis M, et al. *Intensive Care Med*, 2013) The lack of specific criteria for ICU admission decisions has been recently pointed out. (Dahine J, et al. *Critical Care Med*, 2020) Whereas a decision supporting framework was developed to address the issue of limiting or not life-sustaining treatments in the ICU,(Kerckhoffs, *CCM* 2020) no such framework exists for triage.

p.7, lines 33-37: As significant knowledge gaps remain about the provision of potentially nonbeneficial care, more studies on the topic have been called for (Nates JL, et al. *Crit Care Med* 2016; doi: 10.1097/CCM.0000000000001856). Triage is an important area in this respect. Numerous patient-, and context-related factors were shown to influence the decision to admit or not a patient to the ICU, but data are lacking about how these various factors come into play within the decision making process. (Gopalan PD, et al. *J Crit Care*, 2019)

p.7, lines : We aimed to determine which factors physicians consider when they are faced with the ethical issue of providing potentially nonbeneficial intensive care to a critically ill patient, and how these factors influence ICU admission decisions.

2) The result part is not informative enough for an in-depth interview, even for a qualitative study. More information should provide, such as: Is there any reason why the physician role of internists and ICU physicians were different? Chief residents and attendings in ICU group have longer training duration than certified chief residents and residents in internists participants.

Participants' characteristics reflect the structure of the medical staff and clinical working organization in our institution. ICU participants are older and more experienced than internist participants for 2 reasons: 1) most of them train, and sometimes get certified, in another discipline (e.g. internal medicine, anaesthesiology) before training in critical care medicine 2) in our institution, senior ICU physicians, but not residents working in the ICU, are involved in admission decisions on the wards. The internists involved in ICU admission decisions on the wards are chief residents and residents. The latter are especially concerned during night shifts, when the chief residents are not on site. We included ICU physicians and internal medicine physicians so as the participant sample was representative of the physicians routinely involved in ICU admission decisions in our hospital. ICU consultations for critically ill inpatients are requested by the internist in charge after he assessed the patient and if he deems it appropriate. This pre-triage phase is illustrated in Figure 1. "Triage to intensive care: decision making scenarios" of the revised version of the manuscript. During triage, the internal medicine physician and the ICU physician collaborate to make the final admission decision, and they have different roles. We reported data about physicians' roles during triage in a previous publication (Cullati S, et al. *Internists' and intensivists' roles in intensive care admission decisions: a qualitative study*, *BMC Health Serv Res*. 2018; doi: 10.1186/s12913-018-3438-6). Internal medicine and ICU physicians have a shared perception of their practical roles during triage. Internists are expected to recognize the symptoms and signs of severe acute illness, to call the ICU physician at the right moment, i.e. not too early and not too late, and to give him the relevant clinical information and the patient's goals of care. ICU physicians are expected to personally assess a critically ill patient on the ward (no triage over the phone), to give expert advice and make quick decisions, and to manage the access to the ICU.

We gave more information about the roles of the physicians during triage in the Methods section and commented about the participants' characteristics in the Results section.

p.9, lines : We included physicians from the two specialties because triage is a collaborative process in our institution. The internal medicine physician gives the ICU physician the relevant clinical information and goals of care of the critically ill patient. The ICU physician personally evaluates the patient and gives expert advice. The two physicians discuss together about admitting or not the patient to intensive care, but the ICU physician usually has the final say about it.

p. 9, lines : Participant sample was representative of the physicians making ICU admission decisions in our institution.

p.12, lines : Participants' characteristics reflected medical staff's training background and working organization in our institution. Most ICU physicians train in a primary specialty before training in critical care medicine. Since only senior ICU physicians evaluate critically ill patients on the wards, the differences in age and experience between intensivists and internists participants were expected.

3) Pls compare your findings to the mainstream and latest studies relevant to your research. I have suggested some ref. and pls make further and focused search.

We discussed our findings in relation with other studies more in detail, and added some references.

Since the aim of our study was to determine how physicians solved the issue of ICU admission decision when they felt caught in an ethical dilemma regarding the provision of potentially nonbeneficial intensive care, we kept the focus of the discussion on this topic (see also our response to comment 1).

Reviewer: 3

Dr. Alexandra Papaioannou, University of Crete

It is an interesting study, addressing issues all ICU doctors face.

1) However, in methods, (and since it is a secondary analysis) it is not clear whether the results presented here reflect only the cases where the participants considered that ICU admission was futile. The reviewer is right and we thank her for this comment. The main results of this study reflect the cases when participants explicitly raised the issue of potentially inappropriate intensive care, and discussed it as an integral part of the decision making process for admitting or not a patient to the ICU. These cases could be the ones initially chosen by the participants, but the participants also freely referred to other clinical situations during the interviews. The thematic analysis of the data allowed to identify the ethical dilemma the physicians sometimes experienced regarding the provision of potentially nonbeneficial intensive care interventions.

We clarified this point in the Methods section.

p.9, lines : During the interviews the participants sometimes freely referred to other clinical situations, either to make their point or to expand on the idea they were developing.

p.10, lines : Analysis aimed to identify factors that influenced participants' decision making around ICU admission. In particular, we were interested in understanding participants' views regarding potentially nonbeneficial intensive care ("medical futility").

2) Participants were asked to reflect on 2 patients but it is not clear how many those wouldn't benefit from ICU treatment

The main objective of the study was to explore the decision making process for ICU admissions and its determinants. Hence participants were asked to choose 2 cases in which they were personally involved, and which they found significant as to the way the decision making proceeded. We clarified that the decision making could have gone smoothly or not, that it did not matter whether the patient was admitted or not, and that the clinical situations could be simple or complex. Interestingly, participants described at length situations when the decision was complex because it concerned critically ill patients with advanced disease for whom it was difficult to ascertain whether they would benefit or not from intensive care.

We explained the instructions given to the participants in more details in the Methods section.

p.9, lines : They were asked to choose significant cases with regard to the way the decision was made. We indicated that the decision itself – admission or no admission – was not important, that the decision making process could have gone either smoothly or not, and that the clinical situations could be simple or complex.

VERSION 2 – REVIEW

REVIEWER	Oczkowski, Simon
----------	------------------

	McMaster University, Medicine (Critical Care)
REVIEW RETURNED	11-Mar-2021

GENERAL COMMENTS	Thank you for the opportunity to review this revised manuscript--the authors have addressed my concerns and suggestions nicely, as well as those of the other reviewers. I have no further suggestions or comments.
---

REVIEWER	Liu, Yong Shenzhen hospital,Southern medical university , ICU
REVIEW RETURNED	28-Mar-2021

GENERAL COMMENTS	This article has been revised based on the reviewers' comments and has significant improvement.
---